# Measurements of charm production in pp collisions with ALICE

**Tiantian Cheng[1,2]⋆ for the ALICE Collaboration**

**1** Central China Normal University, Wuhan, China
**2** GSI Helmholtzzentrum für Schwerionenforschung, Darmstadt, Germany

⋆ tiantian.cheng@cern.ch

## Abstract

The production of charm-hadrons at midrapidity in pp collisions at $\sqrt{s} = 5.02$ and 13 TeV was recently measured by the ALICE Collaboration. The results show that the baryon-to-meson yield ratios differ from predictions where the hadronization has been tuned to measurements obtained in $e^+e^-$ and ep collisions for different charm-baryon species. These observations suggest that the charm fragmentation fractions are not universal and that the baryon-to-meson ratio depends on the collision systems. The results are compared to the predictions from Monte Carlo event generators and theoretical calculations.

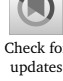
## 1 Introduction

Measurements of the production of heavy-flavour hadrons in high-energy hadronic collisions provide important tests of Quantum Chromodynamics (QCD). The production cross sections of heavy-flavour hadrons can be calculated with the convolution of three terms: the parton distribution functions (PDFs) of the incoming protons, the partonic cross section, and the fragmentation functions describing the non-perturbative transition from charm quarks into charm hadrons. It is generally assumed that the fragmentation fractions are universal regardless of the collision system and energy. However, recent measurements of charm-baryon production at midrapidity in pp collisions showed an enhancement of the yield of $\Lambda_c^0$, $\Xi_c^0$, $\Sigma_c^{0,++}$, $BR \times \Omega_c^0$ baryon to $D^0$ yield ratios with respect to predictions based on $e^+e^-$ and ep experiments, suggesting that the charm fragmentation fractions are not universal among different collision systems.

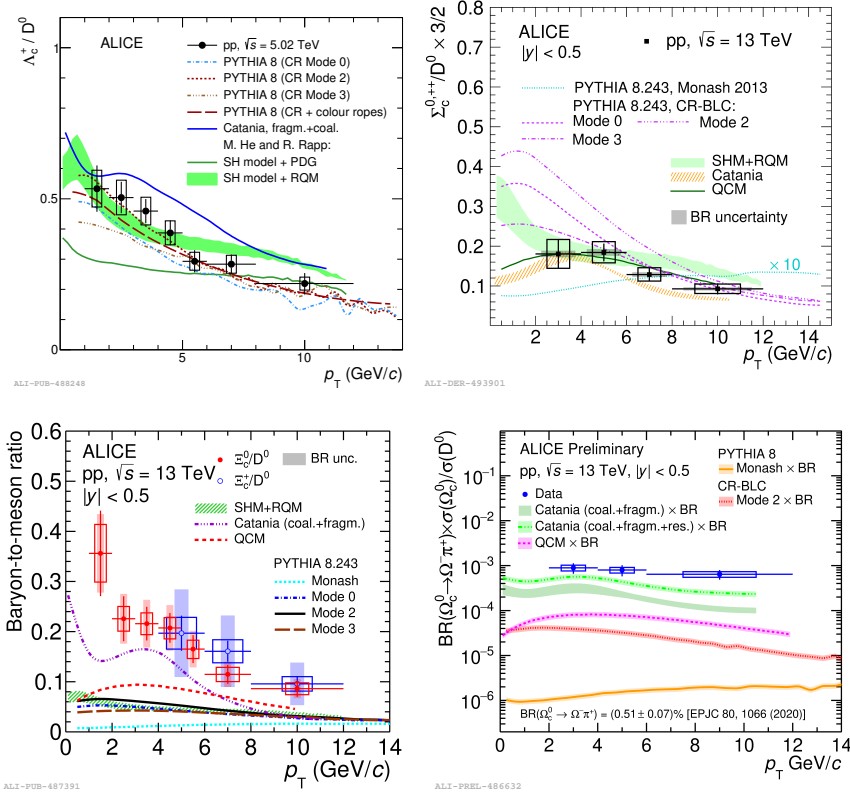

**Figure 1:** The results of charm baryon-to-meson ratios are compared with different theoretical predictions. Top-left: The $\Lambda_c^+/D^0$ ratio measured in pp collisions at $\sqrt{s} = 5.02$ TeV [2]. Top-right: $\Sigma_c^{0,++,+}/D^0$ in pp collisions at $\sqrt{s} = 13$ TeV [3]. Bottom-left: $\Xi_c^0/D^0$ and $\Xi_c^+/D^0$ ratios at a function of $p_T$ in pp collisions at $\sqrt{s} = 13$ TeV [4]. Bottom-right: $BR \times \Omega_c^0/D^0$ ratios at a function of $p_T$ in pp collisions at $\sqrt{s} = 13$ TeV.

## 2 Charm baryon-to-meson yield ratios in pp collisions

Recently, the ALICE collaboration reported measurements of the ground-state charm hadrons, i.e. charm-meson ($D^0, D^+, D_s^+, D^{*+}$ [1]) and charm-baryon ($\Lambda_c^+$ [2], $\Sigma_c^{0,++}$ [3], $\Xi_c^{0,+}$ [4, 5], and $\Omega_c^0$). Both the prompt and non-prompt meson-to-meson yield ratios are well described with the pQCD calculations by using fragmentation functions measured in $e^+e^-$ collisions. However, the calculations significantly underestimate all the measured baryon-to-meson yield ratios. The top left panel of Fig. 1 shows the $\Lambda_c^+/D^0$ ratio in pp collisions, where a clear $p_T$ dependence is observed. It is compared with model calculations. The measurements of heavier charm baryons of $\Sigma_c^{0,++}$, $\Xi_c^{0,+}$ and $\Omega_c^0$ give further important constraints on charm-quark hadronization models. The absolute decay branching ratio (BR) of $\Omega_c^0 \to \Omega^- \pi^+$ is not measured, hence only the BR multiplied to the cross section of $\Omega_c^0$ over cross section of $D^0$ is reported. All the models underestimate the $\Xi_c^{0,+}/D^0$ and $BR \times \Omega_c^0/D^0$ ratios, except the Catania model [6], which implements a new possible scenario for pp collisions allowing low-$p_T$ charm quarks to hadronize via coalescence also in addition to the fragmentation mechanism.

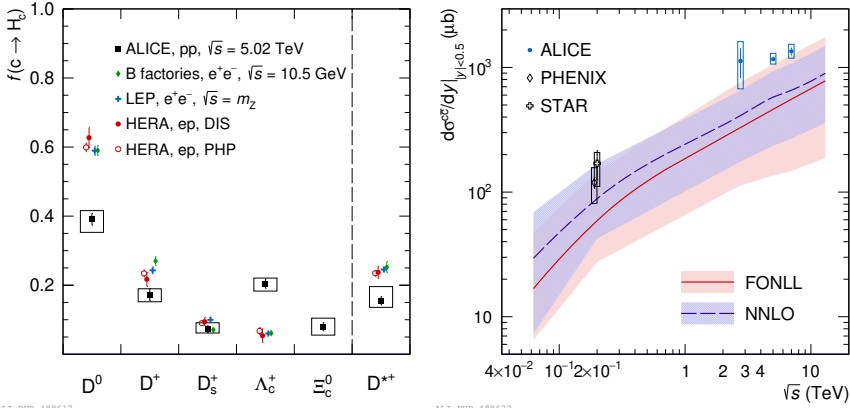

Figure 2: Left: Charm-quark fragmentation fractions into charmed hadrons measured in pp collisions at $\sqrt{s}$ = 5.02 TeV in comparison with experimental measurements performed in $e^+e^-$ collisions at LEP and B factories, and in ep collisions at HERA. Right: Charm production cross section at midrapidity per unit of rapidity as a function of the collisions energy [7].

## 3 Charm production and fragmentation in pp collisions

The charm fragmentation fractions, $f(c \to H_c)$ shown in the left panel of Fig. 2, represent the probabilities of a charm quark to hadronise into a given charm hadron. They were computed for the first time in pp collisions at the LHC using measurements of charm baryons at midrapidity and are observed to be different from the ones performed in $e^+e^-$ collisions at LEP and B factories as well as in ep collisions, providing evidence that the assumption of universality (collision-system independence) of the parton-to-hadron fragmentation is not valid [7]. In the right panel of Fig. 2, the $c\bar{c}$ production cross section per unit of rapidity at midrapidity in pp collisions is calculated by summing the $p_T$-integrated cross sections of all measured ground-state charm hadrons, and is compared with FONLL [8] and NNLO [9, 10] predictions as a function of the collision energy. The $c\bar{c}$ cross sections measured at midrapidity at the LHC lie at the upper edge of the theoretical pQCD calculations.

## 4 Conclusion

Recent measurements of charm-baryon production give stringent constraints to theoretical calculations. The baryon-to-meson ratios are significantly higher at low to intermediate $p_T$ with respect to measurements done at the electron-positron and electron-proton colliders. This observation may indicate that the charm fragmentation functions are not universal for the different collision systems.

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
