# Peer review of "Measurements of charm production in pp collisions with ALICE"

_SciPost Physics Proceedings, doi:SciPost Phys. Proc. 10, 030 (2022)_

## Round 1 · Referee Report · Anonymous (Referee 1) · 2022-1-31

Strengths

A well-presented report, giving a succinct summary of this interesting area and inviting investigation of the original papers.

Weaknesses

The abstract states that "the baryon-to-meson yield ratios are significantly higher than those measured in e+e− collisions for different charm-baryon species", but this isn't shown in the write-up: all we see of the ratios is the pp data against models (getting worse for heavier baryons unless a coalescence mechanism is included -- interesting/important).

The comparison to e+e- and ep is in terms of the yields, not the ratios, where D0 production decreases and Lambda increases... but this isn't highlighted in the text, and doesn't seem as general a conclusion as given in the abstract. Any clarification that can be added, to give a more coherent picture?

Report

The acceptance criteria are met, but if the author is willing to add some small explanation of the issues raised above, it would improve the clarity.

Requested changes

  1. Match abstract/conclusions to the three results shown.

  2. Explain the combination of baryon/D0 ratios and frag function changes in a coherent picture, i.e. how general is the statement that baryon/D0 are generally higher than in e+e- and ep machines?

---

## Editorial Decision

published